# A Cluster Analysis of the Acceptance of a Contact Tracing App—The Identification of Profiles for the Italian Immuni Contact Tracing App

**DOI:** 10.3390/healthcare10050888

**Published:** 2022-05-11

**Authors:** Marcello Sarini, Rossana Actis Grosso, Maria Elena Magrin, Silvia Mari, Nadia Olivero, Giulia Paganin, Silvia Simbula

**Affiliations:** 1Department of Psychology, University of Milano-Bicocca, Piazza Ateneo Nuovo 1, 20126 Milano, Italy; rossana.actis@unimib.it (R.A.G.); mariaelena.magrin@unimib.it (M.E.M.); silvia.mari@unimib.it (S.M.); g.paganin@campus.unimib.it (G.P.); silvia.simbula@unimib.it (S.S.); 2Bicocca Center for Applied Psychology (BiCApP), University of Milano-Bicocca, Piazza Ateneo Nuovo 1, 20126 Milano, Italy; nadia.olivero@unimib.it; 3Mind and Behaviour Technological Center (MiBTec), University of Milano-Bicocca, Piazza Ateneo Nuovo 1, 20126 Milano, Italy; 4Department of Business and Law, University of Milano-Bicocca, Piazza Ateneo Nuovo 1, 20126 Milano, Italy

**Keywords:** technology acceptance model, contact tracing, observational study, cluster analysis

## Abstract

Digital contact tracing apps have been introduced by governments as a strategy to limit the spread of the COVID-19 pandemic. Digital contact tracking is an alternative to traditional contact tracing performed by human tracers who have to reconstruct each contact an infected person had in the recent past by means of interviews. The Italian government proposed the Immuni digital contact tracking app as a solution. Immuni uses Bluetooth technology to anonymously register all close contacts a person had: if she tests positive for COVID-19 then all registered contacts are notified. The main aim of the paper is to propose a cluster analysis of some factors concerning the possible acceptance of the Immuni app to build behaviour profiles that explain and predict the possible behaviours of the respondents. The factors considered referred to three different pillars: the technological pillar, investigated by considering factors from the technology acceptance models family; the health pillar, where variables of the health belief model were used; and the sociopolitical pillar, where some values of the respondents were considered as possible barriers to or facilitators of the acceptance of this technology. As a result of the cluster analysis, three behavioural profiles were built: the ProApp profile, the Hesitant profile, and the AntiApp profile. The first is the profile grouping the respondents who intend to use the contact tracing app; the second is more about people who are favourable of the use of the app, but some issues such as privacy reduce the strength of their intention; the last profile is about people who are less favourable to use the app. We are confident that the behaviour profiles found would be useful to build more tailored communication campaigns to help promote the use of the app by managing factors that could either be facilitators or barriers.

## 1. Introduction

Coronavirus disease 2019 (COVID-19) is a contagious disease caused by severe acute respiratory syndrome coronavirus 2 (SARS-CoV-2). From the first known case identified in Wuhan, China, in December 2019, the disease has spread to become an ongoing worldwide pandemic.

As it is mainly a respiratory disease, the transmission of COVID-19 principally occurs when people enter into contact with airborne particles exhaled by an infected person. This has mainly led to the realization that the risk of infection is highest when people stay in close proximity for a long time, especially in crowded indoor spaces (such as in supermarkets or while using public transport services). Another important characteristic of the virus that led to its increased contagion rate is that people who are infected can transmit the virus to another person up to two days before they themselves show symptoms, and people remain infectious for approximately fourteen days after the onset of symptoms. Consequently, governments identified several preventive measures to try reducing the contagion rate, such as physical or social distancing, quarantining, ventilation of indoor spaces, and the use of face masks and frequent hand washing (see, for instance, [1]).

Additionally, one of the most common preventive measures used to control the spread of COVID-19 is associated with the three T strategy: test, trace, and treatment. This strategy was first introduced by the WHO in 2012 [2] for limiting the spread of malaria and focused on diagnostic testing, treatment, and surveillance, i.e., every person suspected of malaria should be tested as soon as possible, every confirmed case should be treated with the most effective antimalarial drug, and the disease should be tracked through a timely and accurate surveillance system.

A principal aspect of making the three T strategy effective is indeed tracing, the activity of identifying who had contact with an infected person as soon as that person tests positive. Obviously, a positive person has to be properly treated, but the critical issue for limiting the spread of the contagion is that all of the people who were in contact with them have to be warned and tested as soon as possible to avoid an exponential growth of the contagion curve.

Traditional contact tracing is performed by teams of tracers who have to rebuild all the interactions a positive person had. This process consumes both time and resources. It consumes resources because many well-trained professionals are needed to interview people to rebuild chains of contact and to identify the potential dangers for people who were in contact with infected people (and this is especially true in times of high contagion rates). It is also time-consuming because rebuilding the chain of contacts requires a great deal of time; evidently, the time needed to identify contacts is a critical issue, as timeliness is a must for the three T strategy to work. Another possible way of solving the issue of having faster and cheaper contact tracing was introduced after a few months of the worldwide spread of the pandemic. This was the implementation of digital contact tracing solutions, where the human tracers were either replaced or supported by smartphone apps, keeping track of all interpersonal contacts.

This was also the case in Italy, where the Immuni app was proposed as the only official digital contact tracing app. Immuni [3] preserves a user’s privacy by storing locally, in their smartphone, a list of all the temporary exposure keys of the people using the app with whom they entered in close proximity with for long enough. The temporary keys are exchanged leveraging the Apple framework, the Google Exposure Notification framework, and the Huawei Contact Shield framework, which are based on the Bluetooth low-energy system of the smartphone. When a user tests positive for SARS-CoV-2, the user has the option to upload to a server their recent temporary exposure keys. This operation can only happen with the validation of a healthcare operator related to the Regional Healthcare Service (Servizio Sanitario Regionale). The app periodically downloads the new temporary exposure keys and uses them to derive the infected users’ proximity identifiers for the recent past. It then matches the identifiers against those stored in the smartphone memory and notifies the user if a risky exposure has occurred.

However, digital contact tracing, though useful, could raise many potential issues because people are not used to being traced or at least not being explicitly traced. In reality, many apps require both gathering and storing much more information than a contact tracing app in order to provide their services, but this does not seem to raise the same amount of criticism. Unfortunately, as also demonstrated by preliminary research [4], a contact tracing app requires approximately 60% of the overall population to download and use it to be considered effective in its work. Consequently, any possible issue, such as privacy, which could limit the widespread use of the contact tracing app, is critical in this context.

The answer for the first issue (the lack of privacy) was mainly addressed from a technological standpoint. In fact, to reduce its impact, most of the technologies chosen by governments made use of the so-called decentralized models [4]; by using these models, there is no a central repository where the data about contacts are stored. Instead, each app keeps the users’ data locally on the associated smartphone, so that, in theory, no one else can access that data. In this way, privacy could be preserved. However, a technological solution to preserve privacy in these cases would not be enough. In fact, it is not always obvious to users that these kinds of solutions are respecting their privacy (as investigated in [5]). This aspect will be investigated by our study.

To overcome the second issue (to have at least 60% of the population download and actively use the contact tracing app), the strategy considered by governments was to promote proper communication campaigns regarding the use of this app.

Based on these considerations, a central issue of the study is to identify the different meaningful factors that could contribute to either the success or the failure of such a technology. The use of a smartphone application to trace those a person was in contact with in order to prevent the spread of COVID-19 is truly a novel situation, so we need to start from known research but then translate those conclusions and apply them to this unknown context. For this reason, we had several long meetings, and at the end, each of the authors proposed some constructs which could be useful for the context based on their own knowledge. The constructs considered have been organized along three main pillars which we felt could have consequences for the behaviours of the potential users of the Immuni app. The first pillar was technology, which focused mainly around the family of technology acceptance models, to identify the intention of using the app, given some beliefs and attitudes towards apps in general. The second pillar related to the health aspects, as we considered that the use of the app could be influenced by how people perceive the danger of the COVID-19 pandemic. The last pillar is more related to values and motivation that could drive people to accept the app given some limitations related to the use of Immuni, such as privacy concerns. Due to the novelty of the situation, in which very different aspects were combined across the three pillars, we decided to adopt a more explorative approach, as we did not have a strong demonstrated theory about how to combine these different aspects to understand their influence in promoting or not promoting the use of the app. In this way, we are approaching exploratory data analysis as a ”philosophical and artistic approach to gauge every nuance from the data at an early stage” [6], to establish a "foundation” [7] for the interpretation of our data.

Accordingly, we decided to apply cluster analysis to build some behaviour profiles concerning the acceptance of a digital contact tracing app. This reflects our intent to use an ”unsupervised classification” method to ”assign instances to classes that are not defined a priori and that are (usually) supposed to somehow reflect the underlying structure of the entities that the data represent” [8]. In particular, we focused on the opportunity to discover some identikits for the different people involved in the possible use of the Immuni app to orient institutions to build effective communication campaigns to persuade people to use these solutions, or to anticipate the failure of such a solution and to invest in other possibilities.

## 2. Theoretical Background

There are several studies considering the criticalities and factors of success of the use of contact tracing apps to limit the spread of the pandemic in different countries. To our knowledge, none of these studies covered all three pillars. Some of them focused on the health belief model [9], as we also did in our study; as for this specific situation, health issues can have an impact on the possible use of the Immuni app. Others focused mainly on the more traditional family of technology acceptance models [10], and we also considered this aspect, as any introduction of the voluntary use of a new technology could be effectively investigated by taking these models into account. Therefore, we decided to combine the health belief model with a family of technology acceptance models, as other researchers did in previous studies [11]. A few studies about contact tracing apps investigated a broad set of values behind the possible choices of the respondents and their intentions to use a contact tracing app. Some of them focused solely on perceived threats, lack of control, and ideological beliefs [12]. Other focused on privacy concerns [5]. However, most studies on the use of a digital contact tracing solution mainly focus on one of the following aspects: health issues, acceptance of the technology, or political values. We tried to build a more comprehensive model by considering all of these aspects together. In fact, we considered three pillars as the constituents of our model of investigation: the technological aspects, the health aspects, and the value aspects. It will become clear how the technological pillar and the health pillar are intertwined, as we are focusing on the mutual influence of the two; in particular, we will be reflecting on how the use of technology is influenced by factors related to the perception of health.

### 2.1. Technology Acceptance Model and Health Belief Model

The TAM [13] was built on a consolidated theoretical model similar to the theory of reasoned action [14,15]. The TAM key variables received a lot of empirical support: perceived usefulness (PU) was defined as the degree to which a person believes that using a given system can improve their work performance; perceived ease of use (PEOU) was defined as the degree to which the user believes that using a specific tool will be simple. Finally, the degree to which a person intends to use a particular technology is characterized as (behavioural) intention to use (BI) [16]. The TAM asserts that PU and PEOU can result in acceptance, which has a direct influence on BI. In addition, according to the TAM, perceived utility is determined by the simplicity of use. Given the same features and functionalities, the more user-friendly a technology is, the more useful it is considered [17]. This concise theoretical model [18] provides a framework for evaluating how different factors may influence a person’s usage and acceptability across disciplines [19]. Although the TAM has been frequently utilized to describe health-technology-related behaviour, its impact on digital tools use for health-related purposes can only be completely understood by including components of the HMB that explain people’s health beliefs [11,20]. One of the first models developed to understand how the choice to engage in health-related behaviours develops is the health belief model (HBM) [21]. The HBM is a model of change developed to explain and predict health-related behaviour, which relates psychological theories of decision making that attempt to explain actions in a decision-making situation to an individual’s decision about health behaviour. The health belief model hypothesizes that health-related action depends on the simultaneous occurrence of several classes of factors: the existence of sufficient motivation (or health concern)—to make health problems salient or relevant, the belief that one is susceptible or vulnerable to a serious health problem or the consequences of such a condition must be present (often referred to as perceived threat); in addition, the belief that taking a particular beneficial action would reduce such susceptibility or threat must be present. In particular, perceived susceptibility refers to the subjective assessment of the risk of developing a health problem; the model predicts that those who feel susceptible to a given health problem will engage in behaviours to reduce the risk of developing the problem and in behaviours to promote health, whereas those who believe they are at low risk will be more likely to engage in unhealthy behaviours and thus develop a disease [21]. Perceived severity refers to the subjective assessment of the severity of a health problem, including its potential consequences; in particular, it regards the assessment of the gravity of the disease, the possible consequences, such as disability or pain, and the impact on social and work functioning [22]. Indeed, if a person believes that an illness is not medically serious but may induce serious financial consequences due to absence from work, then that person may still perceive the illness as highly serious. In the same way as susceptibility, the HBM predicts that individuals who perceive a given problem as serious are more likely to engage in behaviours to prevent the problem and reduce its severity, whereas those who do not perceive the problem to be serious will not engage in such actions [21]. Another factor that promotes health-seeking behaviour is health consciousness, described as ”the degree to which health topics are integrated into a person’s daily events” [23]. People who are health-conscious are aware of and worried about their well-being, which motivates them to improve and/or preserve their health. Over time, the HBM has continued to be an important reference point for explaining and predicting the acceptance of medical and health recommendations, extending the use of sociopsychological variables to explain preventive health behaviour, such that the model has been applied to the prediction of an extraordinarily wide range of health behaviours among a variety of populations. It is possible to define some limitations of the HBM. Subsequent research has attempted to overcome these limitations, integrating the HMB with the technology acceptance model (TAM).

### 2.2. Computer Self Efficacy

In the context of technology, self-efficacy is expressed in the form of an awareness of one’s ability to harness one’s resources and their capacity to use a technological device in a positive way. People are therefore influenced by the perception of their own self-efficacy in their acceptance and use of technologies, complementing the contributions of the TAM variables, which are already fully accepted in the literature [24,25,26,27].

Several studies have emphasized that computer self-efficacy (CSE) is an individual’s judgement of the respondent about their confidence to be able to use a computer, which is an antecedent of the perceived ease of use of the TAM model. Therefore, the more a person believes themselves to capable of being able to use a technology, the less the expected effort will be to use it [28,29]. This could, in principle, also positively correlate in the TAM model with the intention to use that technology. Thus, consistent with the HBM and TAM theories, for a change to be successful, people must have an incentive to act by believing in their own abilities, they must perceive themselves as competent, they must feel threatened by their current maladaptive behavioural patterns, and they must believe in the possibility of change and that it can be beneficial—thus having confidence in the tools they use. This would mean, in the context of our study, that a higher computer self-efficacy would imply a higher intention to use Immuni.

### 2.3. Moral Disengagement

Moral disengagement (MD) is a multidimensional construct that comprises eight mechanisms of moral control deactivation people use when they engage in immoral behaviours that conflict with their values [30,31]. People who decide against the adoption of the Immuni app disengage from the moral obligation of altruistic behaviours. People tend to justify their moral transgressions according to the following cognitive mechanisms: (1) moral justification (MJ)—based on the appeal to superior aims, which camouflages the reprehensibility of the action; (2) euphemistic labelling (EL)—based on the use of language to sanitise actions and often reflecting the role of media communication in the shaping of public opinion; (3) advantageous comparison (AC)—based on a comparison with more objectionable actions in order to diminish the seriousness of the action performed; (4) distortion of consequences (DC)—based on the negation of the harm and damage caused by the action; (5) diffusion of responsibility (DOR)—based on diffusing responsibility to everyone in order to obscure personal accountability; (6) attribution of blame or guilt (AOB)—when offenders characterise themselves as victims and denigrate victims as being themselves responsible; (7) dehumanisation—when offenders devalue victims of maltreatment as not human, which is not considered in this context. Previous research has shown the predictive role of moral disengagement on various unmoral behaviours, including illegal downloading [32]. In our study, we attempt to verify the mechanisms of moral control deactivation which may explain low intention to adopt the Immuni App.

In general, it is expected that highest values of moral disengagement would be associated with a weaker intention of using the contact tracing app. Further, the identification of the moral disengagement mechanism that can predict adoption avoidance will help the design of communication strategies for the promotion of the app.

### 2.4. Institutional Trust, Political Orientation, Core Political Values, and Prosocial Responsibility

The novelty of this study is in our attempt to combine different domains to provide an overall vision of those behaviours which could either favour or not favour the intention to use a digital contact tracing app. Our idea is that the use of the Immuni app can also be influenced by sociopolitical attitudes and ideologies that may shape individual behaviour. Political ideology may refer to beliefs concerning how government or society should function [33]. Most ways of depicting these principles are organized around the left–right-dimensional political values or liberalism–conservatism. People tend to use this dimension to place and orient themselves in the political realm and reduce its complexity [34]. Research has shown that political ideology can be used to understand times of crisis [35] and the COVID-19 pandemic as well, with conservatism reducing compliant behaviour [36]. Nevertheless, some researchers have enlarged this unidimensional vision to include personal values as important organizers of political attitudes [37]. Specifically, core political values are general normative assumptions about government, citizenship, and society [38]. Schwartz et al. [37] considered a more enlarged range of core political values. Here, we focused on those that we speculated might have a role in determining the adherence to the use of the Immuni app. The first political value is law and order (LO), which refers to the enforcement and obedience of the law’s protection against threats to the social order. The second political value included here is civil liberties (CL), reflecting freedom for everyone to act and think as they consider most appropriate. Another relevant sociopolitical attitude is institutional trust (IT), which refers to the belief that the institutions will act with benign intent, common interest, and fairness for the citizens’ good [39]. Trust is essential in society to foster the provision of public services and democratic practices of cooperation and compliance without relying on coercion. During a pandemic, uncertainties increase and may undermine trust [40], crucial to maintaining public order. As ample research has demonstrated so far, institutional distrust can hinder the success of policymaking [41] by reducing, for instance, negative perceptions of the virus and compliance behaviours for measuring and counteracting the disease [36].

Some recent research has focused attention on the responsibility to protect others as a key driver of prosocial and civic behaviour. Furthermore, in the context of the pandemic, authorities have appealed to awareness of prosocial responsibility to motivate people to adhere to COVID-19 countermeasures [42]. Compliance with these measures is crucial in fighting the pandemic. Prosocial responsibility (PR)—conceived as the awareness that one’s behaviour influences others’ health—has been identified as an important antecedent for respecting measures to counteract the spread of COVID-19. For instance, research has shown that prosocial responsibility influenced a higher frequency of hygienic practices and less hoarding in response to the pandemic [43]. Bobowik and colleagues [44] found that prosocial responsibility was related not only to compliance with hygienic practices and maintenance of social distancing but also to support for diverse policies aimed at fighting COVID-19, such as the acceptance of restrictive policies including the limitation of movement. Interestingly, Kokkoris and Kamleitner [45] confirmed that prosocial responsibility predicted compliance with self-isolation and social distancing measures and acceptance of digital surveillance for the sake of public health. In line with this research, we speculate that the use of the Immuni app can also be motivated by prosocial responsibility—the willingness to accept increased digital surveillance for the sake of public health, at the detriment of privacy.

### 2.5. Privacy

Privacy (P) concerns are often a critical aspect for determining the success of a certain technology. In particular, this is more relevant when the data related to the technology access very sensitive data, such as health conditions. Here, the privacy aspect is far more critical as the health condition, being positive for COVID-19, is connected with others (both closer and farther contacts).

In this specific context Privacy represents an issue as it could lead to a conflict for a person who considers at the same time two opposite consequences of privacy: on the one hand, the concept of privacy related to the idea of protecting and preserving freedom at all costs; on the other hand, the responsibility that preserving privacy could facilitate the spread of COVID-19. We considered two different dimensions of privacy: invasion of privacy, when people perceive that they have lost control over their personal information and interactions with others [46,47]; and a more technological vision of privacy, privacy awareness, which relates privacy aspects with app use. We argue that a stronger valuation of privacy leads to a lower intention to use the contact tracing app.

### 2.6. Perceived Locus of Causality

A fundamental element in promoting behaviour is individual motivation, considered both in quantitative (how much the person is motivated) and qualitative (what kind of motivation the person has) terms. According to SDT, a crucial factor for motivational quality is the perceived locus of causality (PLoC) of a certain behaviour [48,49]. Depending on where the PLoC falls in the internal–external axis, motivation moves on a continuum ranging from autonomous to controlled motivation. Autonomous motivation relates to behaviours that are perceived as originating within the person because of the pleasure derived from the behaviour itself or because of the perceived congruence of the behaviour and of its consequences in the person’s system of values and goals. In contrast, controlled motivation relates to behaviours that are not perceived as originating from the individual, but are enacted to comply with external contingencies of pressures. In the case of controlled motivation, behaviour is perceived as heteronymous, and the individual does not feel ownership or responsibility over their actions. Scientific literature [50,51] has underlined the central role of autonomous motivation in promoting health behaviours, changes in lifestyle, and their maintenance in the long term: studies that take intentions along with behaviour into account highlight how autonomous motivation is predictive of both [52,53]. In this view, the possibility of using the Immuni app could be explained in terms of autonomous motivation.

## 3. Materials and Methods

With respect to the cited studies, we decided to propose a different approach as we did not focus on the inferential statistics to relate variables to explain the phenomena. We want to proceed here by taking into account the novelty of the situation represented by the use of a technology for completing contact tracing activities. In this way, we decided to take a less theoretic approach and instead investigate the data and read what they tell us in a more explorative way. In particular, we used an unsupervised classification approach, cluster analysis, which provides an artful and scientific method to receive hints from the data without designating any specific expectation. This approach is not new, since, in other domains, other scholars have applied cluster analysis to identify profiles of the respondents (see, e.g., [54]).

### 3.1. Study Design and Data Collection

The data collection for this study was conducted from 10 July to 18 August 2020, utilizing a web-based survey administered using the Qualtrics tool. A total of 1111 participants agreed to participate in the study after providing informed consent. Some subjects were recruited through social media word-of-mouth communications, and people internal to the Bicocca University (staff, students, and personnel) received an email from the central offices which advertised the study. The questionnaire (Appendix A) included 104 items, and the total time requested of respondents was approximately 20 min.

### 3.2. Ethical Considerations

The study was conducted after receiving ethical approval from the Local Commission of the Department of Psychology for minimal risk studies. Participation in the study was completely voluntary, and detailed information about the study’s objectives and significance were provided to participants in an informed consent form attached to the web-based survey. Personal identifiers, including name and email address, were not collected to preserve anonymity. Only one response per participant was allowed.

### 3.3. Survey Questionnaire

We had long discussions about the preparation of the questionnaire to be used to investigate the feasibility of using the Immuni app as a contact tracing tool. The proposal of a technological solution for this kind of activity posed several issues. It was important to try to consider all the different aspects related to such a situation. On the one hand, undoubtedly, the COVID-19 pandemic brought up many problems about health issues; so, in our opinion, it was necessary to investigate—with a minimal number of variables—the health issues related to COVID-19, because we were strongly convinced that the use of a contact tracing app would be affected by the belief of the user about how severe the pandemic was. Therefore, investigating the health side was considered an important aspect for taking the positive factors that could, in principle, promote the use of the app into account. On the other hand, this kind of solution, especially one that was not only pushed by the government but which could also, in theory, impact matters of privacy (both with regard to personal data, considering that contact tracing could also be used as a way to control the activities of the user, and in terms of health data, as being positive for COVID-19 has both private and social consequences, as we are asked to avoid a possible spread of the contagion). Therefore, these other aspects are more related to privacy and other personal values which could in principle be read as barriers to the use of Immuni. Therefore, the idea is that, at the end, the possible success of a digital solution to contact tracing could only be reached with a very difficult balance between different and sometimes contrasting issues—when does privacy become more valuable than the health of others? When is health to be considered a private issue and when is it a social one, especially in a pandemic situation? When should a government be able to violate the privacy of its citizens, even for a good reason? After long discussions, we selected different variables taking these contrasting issues into account, while also taking the three pillars of this study into account: the technological aspects, i.e., how confident people are when using this kind of technological solution, how they perceive they will be able to use it, and how much they consider this solution to be useful, and so on; the health aspects, i.e., how much they care about their health, how much they are afraid they will be infected by COVID-19, and so on; the value aspects, i.e., how much they consider their privacy and their freedom of choice above all, how much they feel responsible for the wellness of others, how much they think governments should enter and affect their life so directly, and so on.

### 3.4. Database Preparation

When the questionnaire was an active link, 1111 people opened it. However, at the end of the deadline for the completion of the questionnaire, about 500 people had almost fully completed the questionnaire. To glean meaningful analysis from our data, we opted for a conservative approach: we decided not to apply techniques to manage missing data, and we kept only the cases where the respondents fully completed the questionnaire. In this vein, at the end of database preparation, we obtained 408 cases for our analysis.

### 3.5. Tools

To perform our analysis, we combined the use of different tools. From the database saved in .sav format, we used the Jamovi tool mainly for the first analysis concerning the definition of the variables and then the preliminary descriptive statistics. For more complicated analysis, we used the RStudio tool, and we installed several packages: factoextra for cluster analysis; reshape2 and ggplot2 for plotting data; MASS, caret, and nnet for multinomial logistic regression. When an exchange of data was needed between Jamovi and RStudio, we used the jmvconnect library.

In the following sections, we describe the variables we decided to consider, given the motivations described above.

### 3.6. Sociodemographic Data

In addition to the usual sociodemographic data, such as age, gender, and study level, for the sake of the study, respondents were also asked where they spent their life during the lockdown (e.g.,living outside the region). This was considered useful, because in Italy, during the first wave of the pandemic, there were very different situations, from mild to severe, according to the specific regions of residence. For instance, Lombardy was the region with the most severe consequences of the pandemic. So, it could be possible to argue at least in principle that the perception of the severity of COVID-19 consequences would be affected by where a person was living, and this in turn could affect their intention to use a contact tracing app. Other data considered interesting for the study were those concerning technological aspects: number of downloaded apps indicated how confident respondents were with using smartphone apps; privacy awareness, in particular, related to the propensity of the respondents either to install or to refuse to install new apps on their smartphone as a consequence of the permissions asked before downloading them; and which data were considered the most private (data considered private). Another relevant aspect we investigated was the Immuni app: this was because the object of our study was to understand the respondents’ knowledge about the app (knowledge of Immuni), and whether the respondent had actually downloaded it (downloaded Immuni).

### 3.7. Measures

Technology acceptance was assessed using items from previously published works, adapted to the specific context of the current research [13,55]. Perceived usefulness was evaluated through four items (e.g., “Using the Immuni app will help me manage my health”); perceived ease of use was measured using four items (e.g., “Learning to use the Immuni app to get health information will be easy for me”); behavioural Intention to use the app was evaluated via four items (e.g., “I plan to use the Immuni App in the next three months”); social influence was assessed using eight items (e.g., “My general practitioner would approve the use of the Immuni app”); the facilitating conditions were evaluated using three items (e.g., “I have technical know-how (i.e., “the knowledge and operational skills) to use the Immuni App”); finally, attitude was measured via four items (e.g., “I like the idea of using the Immuni App to manage my health in relation to COVID-19”). HBM dimensions were assessed by adapting items used in previous research (e.g., [11]). Perceived severity of chronic disease was evaluated via four items (e.g., “I am afraid of facing attack or deterioration due to COVID-19”); Health consciousness was measured using eleven items (e.g., “I consider myself very health conscious”). Both the TAM and HBM dimensions were assessed using a scale ranging from 1 (strongly disagree) to 5 (strongly agree).

*Computer self-efficacy* was evaluated using 10 items adapted from [56], assessed on a 10-point scale (1 = not at all confident; 10 = totally confident). The scale was adapted to smartphones, the technology considered in our paper (e.g., “I could utilize a smartphone app like the one presented… Someone showed me how to do it earlier”).

*Moral disengagement* was assessed by measuring three items for each of the following mechanisms: moral justification (e.g., “I believe it is unfair that people should share their personal information”); euphemistic labelling (e.g., ”Beyond the Immuni App there must be financial interests”); advantageous comparison (e.g., “The risk associated with COVID is better than the risk associated with the loss of freedom”); distortion of consequences (e.g., ”The Immuni App might cause many social problems”); diffusion of responsibility (e.g., “Sharing own personal information when the others do not do it does not make any sense”); attribution of blame or guilt (“The failure of the Immuni App will be due to the Government incapacity”). The items were adapted from the scale proposed in [32] and were measured with a 7-point scale (1 = strongly disagree; 7 = strongly agree).

*Political orientation* was measured on a single item, asking participants to position themselves on a 10-point scale ranging from 1 (far left) to 10 (extreme right) (see [35]. Concerning the measure of core political values, we adapted three items taken from Schwarz et al. [37] to measure each of the following two values: law and order (e.g., “Political measures to increase security should be promoted at this time, even if it could mean sacrificing the freedom of citizens”), and civil liberties (e.g., “The right to individual freedom is inviolable and has to be maintained at all cost”). Answers were provided on a 7-point scale from 1 = completely disagree to 7 = completely agree. Since trust in specific institutions (institutional trust) was found as determinants of adherence to countermeasures against coronavirus pandemic, we measured trust toward five institutions relevant to the pandemic’s management (e.g., the government, the national health institute). The following question preceded the items: “To what extent do you trust the following institutions?” (This was measured on a 7-point scale (1 = not at all, 7 = very much). We adapted three items from Bobowik et al. [44] to measure prosocial responsibility behaviour, e.g., “I am also as an individual citizen responsible for protecting other citizens from the pandemic”. Answers were provided on a 7-point scale, ranging from 1 = strongly disagree to 7 = strongly agree.

*Privacy* was evaluated via three items, adapting the scale proposed in [57] to the use of Immuni app (e.g., “I feel that the use of Immuni makes it easier to invade my privacy”). Privacy was assessed on a scale ranging from 1 (strongly disagree) to 5 (strongly agree).

*Perceived locus of causality* was measured by eight items adapted from a previous measure [48]. Participants responded to the item “I plan to use the Immuni app because” with four autonomous and four controlled reasons for doing so. Example items assessing autonomous motivation included “it reflects my values” and “it is personally important to me.” Example items assessing controlled motivation included ”because others would disapprove of me if I did not” and ”I would feel guilty if I did not follow the recommendation”. The items were paired with a 7-point scale (1 = strongly disagree, 7 = strongly agree).

The variables we considered in the analysis were selected from the database with the criterion that variables with Cronbach’s Alpha values smaller than 0.7 were dropped. Then, for any of the variable measured by more than four items, we applied exploratory factor analysis (EFA) to split the results into more meaningful factors. The choice of the factors was driven by the values of the root mean square error of approximation (RSMEA) and the Tucker–Lewis index (TLI), which indicate a good division into factors as they get are closer to zero and to one, respectively. The choice of EFA instead of principal component analysis (PCA) was driven by our more data-oriented approach, without considering previous research concerning the usual factors related to the considered variables as in this study. We are focusing on a very novel aspect regarding the use of a specific technology, and we do not want to rely too heavily on previous results. The selection of variables was carried out using the Jamovi tool [58].

### 3.8. Cluster Analysis

Among the available clustering algorithms, we applied the K-means algorithm, since we used a flat structure of our data, to identify different profiles that could be compared on the basis of the patterns that our data exhibited without any initial hypotheses. First of all, to apply the algorithm, it was important to identify the optimal number of clusters for classifying our data. To find the optimal number of clusters, we ran the gap statistic algorithm in R, and we obtained three as the optimal number of clusters (see Figure 1).

We then applied the K-means algorithm on the scaled data with three clusters, and we obtained cluster 1 with 145 cases, cluster 2 with 141 cases, and cluster 3 grouping 122 cases. We also computed some metrics to verify the good fit of the clustering, as described in Table 1. In particular, we focused on a metric: the ratio between the variance between the groups (between_SS) and the total variance in the data (total_SS). In optimal clustering, this ratio is as good as 100% (in a range between 0 and 100%). In fact, in optimal clustering, since the clusters are very different, most of the total variance is explained by the variance between the groups. Of course, since the variance within each group was very small, it would explain only a small fraction of the total variance in the data. In our case, the ratio (between_SS/total_SS) obtained a value of 76.1%, so we concluded that we obtained a good clustering of our data.

Figure 2 displays how the single cases (each of the respondents, identified by their number) are grouped together to form the three clusters.

## 4. Results

### 4.1. Descriptive Statistics

The sample of the 408 respondents had the following characteristics. Of the respondents, 35% were male and 64.2% were female. Regarding age, the youngest respondents were 19 years old, while the oldest were 69 years old. Of the respondents, 10% were aged between 0 and 20 years, 78% were aged between 21 and 40 years, 10% were aged between 41 and 60 years old, and just 2% of the sample was over 61 years. Of the respondents, 35.4% were bachelor students at the University of Milano-Bicocca, while the same percentage were masters students at the same university. Postgraduate students of the University of Milano-Bicocca comprised 9.5% of the sample, and 19.6% of the respondents did not fall into the previous categories (e.g., internal staff, postdoc students, professors, and researchers). Then, 83.5% of the respondents studied and lived in the same region (Lombardy). These data were considered interesting, since Lombardy was the region with the highest damage from the pandemic, and the respondents could have provided different answers depending on whether they lived in their usual home with their family or not. Meanwhile, 94.6% of the respondents knew what the Immuni app is, and among all of the respondents, 48.4% at the time of the answer had already downloaded the Immuni app on their phone, while 51.6% had not.

### 4.2. Building Respondents’ Profiles from Cluster Data

The main goal of this cluster analysis was to help find meaningful profiles related to different behaviours and attitudes towards the Immuni app.

On top of the cluster data, we built three profiles that provide an interpretation layer of the behaviours, attitudes, and beliefs of the respondents towards the use of the Immuni app. To this aim, to identify the differences for the respondents within clusters, we started to run tests to verify the assumptions regarding the possibility of performing an ANOVA one-way analysis. In particular, we checked the normality of the distribution of the variables by carrying out the Shapiro–Wilk test of normality for the variables. As we discovered that the normality of the variables failed (most of the p-values returned by the tests were less than 0.05), we opted for a non-parametric test, the Kruskal–Wallis test. We performed the the Kruskal–Wallis test for each of the variables with respect to the clusters. We thereby found variables in which the differences were not meaningful, and for the other variables, we carried out the pairwise Wilcoxon rank sum test to perform multiple pairwise comparisons between the means of the groups, so to have a more detailed explanation of where differences lay and to what extent they existed.

We dropped six variables—*HCf1*, *HCf2*, *MDf4*, *MDf5*, *LO*, and *CL*—because the differences computed by the Kruskal–Wallis test were not meaningful, so we considered these variables not to be useful in helping us characterize the differences among the different profiles. Figure 3 provides a general idea of our approach by emphasizing where differences occur in the clusters with respect to the considered variables.

We therefore kept the twenty-two variables with meaningful differences and computed the the pairwise Wilcoxon rank sum test for each of them to identify more precisely how these variables explain the different characteristics of the profiles. We do not propose all of the analysis in detail, but in Figure 4 we present a table that summarizes the differences for the variables, where cells in dark brown are related to cases in which the differences for the average value of the variable in the column is not meaningful for the pair of clusters of the corresponding row. The table in Figure 4 is designed to visually support the interpretation of the patterns of the data related to the different clusters to propose the profiles of the respondents. We reported in the first four rows of the table the average values of the variables for all of the respondents, and for the respondents grouped into the three clusters. The following three rows aim at easily communicate where profiles vary focusing on any of the variable: in fact, in these rows any cell of the table describes the difference between the average values of a variable (a column) for any possible pairs of clusters (a specific row). Looking at any of the cell would hence help to interpret the different possible behaviours of the respondents in relation to the use of the Immuni app.

To facilitate the readings of the data, we showed in green the cells that gave a variable in the corresponding column the lowest absolute value, considering all of the values in the rows describing differences between pairs of clusters. In red, we emphasize the opposite concept, when the value of the difference is the highest in absolute value. So, for instance, in the column related to the variable PEOU, where the differences are meaningful for any of the pairs of clusters, the cell corresponding to the 3-1 row (the difference of the average value of this variable in cluster 3 with the value in cluster 1) is highlighted in green, because, for PEOU, this is the lowest absolute value. The cell corresponding to 3-2 is red, as this value represents the highest difference in absolute value. A conceptually similar table, just for statistically relevant data, is shown in Figure 5, but this one uses the scaled data from the previous table in order to have a homogeneous interpretation of the data, excluding the differences related to the scales for the items of the single variables and the different ways to compute variables from their items (e.g., the fact that CSEf1 is related to the sum of the items while the other variables from the average values of their items).

Observing the Table in Figure 4, it is possible to note some patterns in the data with respect to the differences between the averages of the variables in the different pairs of clusters (the same happens in Table in Figure 5, but let us concentrate on just one table at this moment).

Let us speculate on the differences of the clusters by considering the three pillars: the technological pillar, the health pillar, and the value pillar.

For what concerns the technological pillar, we have to consider ninevariables: *PU*, *PEOU*, *SIf1*, *SIf2*, *SIf3*, *A*, *BI*, *CSEf1*, and *CSEf2*. In general, for most of the variables from the technological pillar (7 out of 9), there are meaningful differences for any of the pairs of clusters. The only two variables where differences are not meaningful (PU and SIf2) regard only the comparison of cluster 2 with cluster 1.

Cluster 1 and cluster 2 appear to be the most similar, for two variables (PU and SIf2) differences are not meaningful, and for most of the other variables, there is the smallest difference (five cells in green out of seven).

Cluster 3 mostly differs from cluster 2, with all of the cells having the highest differences for any of the variables. The values in the cells are always negative, and this means that the average of the values is always smaller for respondents in cluster 3 (and for how the ranking of the items are distributed, this represents a less favourable behaviour in regards to the app from the technological point of view).

The health pillar reflects two variables: PSCD and HCf3. In general, in this pillar, the differences between the clusters are less meaningful, with just two variables with meaningful differences, not involving all of the pairs. For this pillar, people from cluster 3 seem to be less interested in the health consequences of the COVID-19 disease; as for both variables, there are meaningful differences with at least one cluster, and with a negative value.

The value pillar includes eleven variables: *MDf1*, *MDf2*, *MDf3*, *ITf1*, *ITf2*, *PO*, *P*, *PR*, *PLoCf1*, *PLoCf2*, and *PLoCf3*.

In general, for the variables of the value pillar, differences are meaningful for any of the pairs of cluster just for PLoC. This make this variable important for identifying the differences in the clusters.

In this pillar, cluster 3 and cluster 1 appear to have less relevant differences (4 variables, the highest number—MDf1, MDf3, P, and PR—with no meaningful differences).

Cluster 2 and cluster 1 seem the most similar, with all of the cells in row depicted in green or dark brown. However, the differences are quite high in value, especially when the value in the cell is negative, to indicate that, for privacy issues and the moral disengagement aspects, there is a relevant higher value of the average in cluster 1, with respect to cluster 2, emphasizing that, for the people of cluster 1, there are some barriers to the use of the app caused by their values.

As for the previous pillars, cluster 3 and cluster 2 appear to be the most different. The interesting aspect is that the values of the differences here are mostly positive in sign: this holds for the moral disengagement factors considered, and for the political orientation factors, where respondents of cluster 3 appear to be more oriented toward the centre in comparison with respondents of the other clusters, which seem more oriented towards the left side of the scale.

### 4.3. Integrating Profiles with Sociodemographic Data

We considered the following categorical variables for this analysis: the gender of the respondents (male, female) grouped by cluster; the age of the respondents (up to 20 years; 21–40 years; 41–60 years; and 61–80 years), according to the different clusters; the study level (bachelors, masters, phd-postdoc, and other) of the respondents in any of the clusters; the living outside the region factor (yes, no) grouped by clusters. As already described above, this variable could be useful to characterize profiles according to the experience of life during the lockdown. Then, we considered some categorical variables about the technological attitudes of the respondents aggregated according to the following clusters: number of downloaded apps (from 0 to 5; from 6 to 10; and more than 10); their privacy awareness (yes, no); their data considered private (profile, health, picture, position, otherdata considered private (profile, health+picture+position, other ). Then, the most important categorical variables to be considered here were employed to differentiate the profiles according to behaviours related to the Immuni app, the goal of this study: knowledge of Immuni (yes, no) and downloaded Immuni (yes, no). For all of these variables, we performed the chi-square test to identify possible meaningful differences among clusters to better build an interpretation for the profiles.

The results from chi-square tests showed that the clusters differed in terms of gender (df = 2, *p*-value = 0.03348), number of downloaded apps (df = 4, *p*-value = 0.03474), privacy awareness (df = 2, *p*-value = 0.005265), data considered private (df = 4, *p*-value = 0.01231), and downloaded Immuni (df = 2, *p*-value = 1.488 × 10^−5^. For the other variables, clusters did not show a statistically significant difference (age, df = 6, *p*-value = 0.4936; study level df = 6, *p*-value = 0.1381; living outside region, df = 2, *p*-value = 0.3009; knowledge of Immuni, df = 2, *p*-value = 0.5903). It is important to note that the smallest p-value obtained from the tests on these variables was related to the variable downloaded Immuni (the value was very close to zero), which is undoubtedly the most important to characterize the behaviours of the respondents associated with the three profiles, as it reflects a respondent’s real intention to use the app, having downloaded it on her smartphone.

For the variables with a meaningful value from the chi-square test, we performed further analysis: we compared the observed values with the expected values, and then we computed the residuals to identify the highest differences between the expected values and the observed values, to determine whether they were positive and/or negative and to determine the cluster they were in, to interpret these values in light of profile building.

For the gender variable, we noted that cluster 1 was where the highest difference occurred, and it was where the number of male respondents was missing with respect to the expected value. It seems that cluster 1 was characterized by a higher number of female respondents than expected. Then, it is interesting to note that all three categorical variables which we associate with technological aspects related to the use of the smartphone obtained a meaningful chi-square test value, or, in other words, we can consider these technological aspects as discriminants for the determination of the three profiles. Concerning the number of downloaded apps factor, the highest differences were found comparing the observed values with the expected values in cluster 1, where people belonging to this cluster had downloaded and installed fewer apps on their smartphone than expected, so it seems that the people of this cluster are less prone to the use of the apps. Another interesting option is for people in cluster 2: which seem to include more real app users than expected. The privacy awareness variable was especially important for investigating whether respondents refused to install an app on their smartphone due the requested permissions. Members of cluster 2 were, in fact, more likely to download apps on their phone irrespective of the permissions requested. This was also in line with the higher number of downloaded apps than expected. People in cluster 2 seemed to be more confident in the use of the technology, in particular with regard to smartphone apps.

The most interesting aspect to note from the last categorical variable associated with technological aspects—data considered private —in line also with the fact that people in cluster 1 were partly characterized by being affected by the privacy issue—was that, for cluster 1, the privacy of the user’s profile was considered to be higher than expected in the observed data. Finally, the last and probably the most interesting categorical variable, downloaded Immuni, reflects the point at which the Immuni app was downloaded and installed by people in the three different clusters. Here, the most relevant differences with respect to the trend were already identified by analysing the other variables with respect to cluster 2 and cluster 3. In cluster 2, which could be associated with a profile of someone confident about the use of Immuni, the observed number of people who downloaded the Immuni app was higher than the expected number. Conversely, for cluster 3, which could be considered to describe the behaviours of respondents sceptical toward Immuni, the number of people who did not download the Immuni app exceeded the expected number. This difference between the observed and expected numbers was higher than the difference computed for people in cluster 2. Thus, the resistance to use of the Immuni app in profile 3 was stronger than the acceptance of using the app in profile 2.

## 5. Discussion

We can now associate both the analysis of the differences between average values of the variables across the clusters and the differences in the expected values of categorical variables to try to better delineate the three profiles. We consider that we built the profiles of the respondents on the basis of the cluster data, i.e., we defined three profiles with the corresponding aggregation of respondents, as represented by the three clusters obtained from the cluster analysis.

From cluster 1, we built the corresponding **hesitant profile**: people belonging to this profile are similar to people in cluster 2, and they do not seem to have strong reasons to reject the app. In theory, they would be likely to use the Immuni app, but some aspects make them avoid using the app: the most relevant concerns are privacy issues (as described by the trends of variable *P*), a lower number of installed apps than expected, and the privacy of the profile, as the data were considered more private than expected.

Cluster 2 is used to define the **pro-app profile**: people from this profile seems to be more favourable toward the use of the Immuni app. They seem to be more confident about the use of technology in general. In particular, they have downloaded a higher number of apps than expected. They are not going to have problems using the Immuni app, and they exhibit the highest computer self-efficacy. In other words, they seem to be the most favourable toward using technology, and this could hold true regardless of the Immuni app specifically, as they seem to be the most prepared to use a new technology like Immuni.

The third profile, which we decided to call the **anti-app profile**, is associated with cluster 3, in which people seem to be more opposed to the use of the Immuni app due to their values. They do not seem to be opposed to the technology at all; in fact, they did not seem to have less than the expected number of apps on their phone, and they did not seem to have too many privacy concerns, but they also did not seem very confident about their ability to use the app.

Now that we have identified the different characteristics of the three profiles with regard to the possibility of using the Immuni app for contact tracing, we want to propose some reflections on how to use the different profiles to try to improve the use of Immuni app. People with hesitant profiles, similar to people of the pro-app profile, are quite favourable toward the use of the app but are wary of privacy issues. So, for people with the hesitant profile, privacy is a barrier, and for them, a campaign must be designed to remove the barrier of privacy whenever possible, for instance, by clearly explaining that the design of the Immuni app considers preserving the privacy of its users. Those with a pro-app profile seem the most favourable toward using the app; however, there are still many of them who had not downloaded the Immuni app yet. Therefore, it would be useful to focus more on promoting the use of the app, maybe making a more detailed distinction between people of these profiles, to understand whether there are some specific aspects that could hamper the use of the app. It is also possible to explain that people who have not downloaded the app yet are still waiting to do this, so they could be encouraged to do so, because, as their profile’s characteristics indicate, they believe that Immuni is a helpful tool for contact tracing. People from the anti-app profile seem to be the hardest to convince of the goodness of Immuni as a solution for contact tracing. On the one hand, a tailored campaign should help to promote the finding of personal motivation to adopt the Immuni app; on the other hand, people from this profile should be convinced about the ease of use of the app and should be helped to improve their computer self-efficacy with regard to the use of Immuni, as they also seem to have some difficulty using it.

Once we performed this work of profile identification, we considered profiles as a tool that can be used in both descriptive and predictive ways. In descriptive ways, as we already considered, we can use profiles to try to explain the possible behaviours of people belonging to any of the profiles as of a sort of an identikit of the members in regards to the use of the Immuni app. There is then a predictive way to use the profile, that is to give a new respondent the profile to which they are most likely to respond. The techniques used for this second use of the profiles will be discussed in the next section.

### Predicting Profile Membership

Given the importance of the profiles we identified, it would be useful to predict, when considering a new respondent, which profile they will associate with (in terms of the related cluster). There are different techniques which can be used to predict the category of a case (in this case, one of the possible three clusters) according to the values exhibited by a set of variables, such as the classification approaches described in [59]. Among the possible techniques, the two techniques mainly used for multi-class classifications are linear discriminant analysis and multinomial logistic regression. As one of the main assumptions to perform linear discriminant analysis, the assumption of normality, i.e., that the distribution of the predictors is normally distributed within the different clusters, is not satisfied [60], so we can not apply this technique for our prediction task. Hence, we applied multinomial logistic regression and we generated three different models: the first is the model built considering all of the twenty eight variables of the study—the full model. Then, we performed multinomial logistic regression with the aim of finding a model that would guarantee similar performances but with a lower number of variables to predict the related respondent’s profile: this would guarantee an economization of our tool, limiting the size of the questionnaire and the time requested of new respondents to complete the survey. For finding a more economic model, among all the possibilities, we considered two strategies: the first is a strategy which could be termed as a knowledge-based strategy, which advocates that it is ”better to use subject matter knowledge” in variable selection [61]. The second is a more traditional strategy based on step-wise model selection using the AIC (Akaike information criteria) [62]. We built two models using the two strategies. The first model, the knowledge-based model, was derived selecting a subset of the twenty-eight variables with the aim of choosing the most meaningful ones to cover all the relevant issues related to the three pillars which drove our study. In particular, for the technological pillar, we selected the BI variable and the two factors related to CSE; for the health pillar, we selected the PSCD variable; and for the value pillar, we selected the variables MDf1, MDf2, MDf3, and OP, and the three factors related to PLoC (eleven variables in total). The second model, the step-wise reduction model, was computed using the R function stepAIC, which takes into account the value of AIC, which is to be minimized at any step of the reduction process to come up with the final set of features. The process aims at simplifying the model, reducing the amount of information loss without impacting its performance too much [62]. The use of stepAIC also removes the multicollinearity if it exists. At the end of the step-wise reduction, we obtained a model with twelve variables: PU, PEOU, SIf3, CSEf1, PSCD, HCf1, MDf3, MDf5, FIf1, FIf2, OP, and SOR.

In Table 2, are reported three performance measures to compare the three models [63]: accuracy, AIC, and LogLoss. Accuracy reflects the percentage of correct predictions using the model: the closest to one, the better the model; AIC quantifies the amount of information loss due to removing variables. If we are given two models, then we will prefer the model with a lower AIC value. LogLoss is an important classification that evaluates how close the prediction probability is to the corresponding real value. LogLoss is a good metric for comparing models. For any given problem, a lower LogLoss value means a better prediction [64]. It is possible to note that full model and the knowledge-based model perform better for some metrics, if not all: the knowledge-based model performs better for factors which concern accuracy, so it is able to predict the right cluster for a given case almost perfectly. Moreover, the knowledge-based model minimizes the AIC value, i.e., the amount of information loss during simplification. For that which concerns LogLoss, the full model is best at minimizing its value, so it minimizes the probability of diverging the prediction from the actual value. The step-wise reduction model does not perform best for any of the metrics, but it scores in the middle of the other two models for any metrics. Given that we are more interested in considering smaller models, it is possible to conclude that both refined models could be safely used to predict the cluster of our respondents. This is an interesting result, showing that the models built with two different approaches perform in a similar way; therefore, it is possible for researchers to choose which of the two models to use, according to the context of use.

## 6. Conclusions

In this study, we investigated which factors could either facilitate or hinder the use of a digital contact tracing technology, the Immuni app, in Italy. To identify these factors in such a novel situation—the use of an app to try to limit the spread of pandemics—we considered three different pillars of influence: the technological pillar, to specify the intention of using the app, given some predictors from the family of technology acceptance models; the health pillar, which focused on identifying factors related to the fear of COVID-19 and its consequences on health as a set of factors possibly influencing the choice of using the Immuni app; the values pillar, to specify the values that could need to be balanced (such as privacy rights) in order to accept the use of the app. Given the novelty of the situation, we decided to adopt a data-driven approach using exploratory data analysis. Specifically, we opted for a cluster analysis to build respondents’ profiles and to use these profiles to promote suitable communication campaigns, and to determine whether the use of a contact tracing app might not seem to be a feasible solution. Starting from the clusters identified, we built on top of them three different profiles: the hesitant profile, the pro-app profile, and the anti-app profile. The hesitant profile is for respondents who are quite favourable toward the use of Immuni app, apart from some distinct aspects, most notably privacy issues, which could hamper their intention of using the app. The pro-app profile is for respondents who are favourable toward and confident in the use of any app, and this especially holds true for the Immuni app. The anti-app profile is for respondents who are reluctant to use the Immuni app, both due to their computer self-efficacy and the values supporting their motivations to use the app. These profiles could be useful for governments to design more effective communication campaigns for promoting the use of the Immuni app targeted to the different potential users. Moreover, the relevance of this study was also to set up an analysis protocol that could be useful to interpret the inner structure of data by means of cluster analysis and other analysis techniques to interpret the differences in the values of the variables between different pairs of clusters.

There were several limitations in the study: one concerns the homogeneity of the sampled population, since respondents were mainly students of Bicocca University. However, the chi-square tests performed on the demographic data did not identify meaningful differences from our sample with respect to age, gender, or study level. Therefore, it is reasonable to argue that we could find similar results with other target populations. Regardless, it would be interesting to present the survey to other targets. Another interesting aspect that emerged about the use of contact tracing apps [65] concerned the effectiveness of their use when targeting specific communities. In this vein, we can consider the design of a contact tracing app specific to the population of students of Milano-Bicocca, also taking into account the results of our study. Other limits to validity could be related to the period of administration of the questionnaire, and the decrease in perception of the level of threat posed by COVID-19. Other limitations could be in the approach used, as we focused just on exploratory data analysis. Future works could use these results for confirmatory analysis.

A final consideration about the Immuni app is that it does no only represent a sociotechnical system; its use is affected by its users and their attitudes, beliefs, and intentions. It is also a political system, as its use is connected to the way in which politics make its effective use possible and feasible. As an example of this, the initial use of Immuni was limited, since the organizational structures managing the overall tracing process were not able to deal with some phases of the process. 

## Figures and Tables

**Figure 1 healthcare-10-00888-f001:**
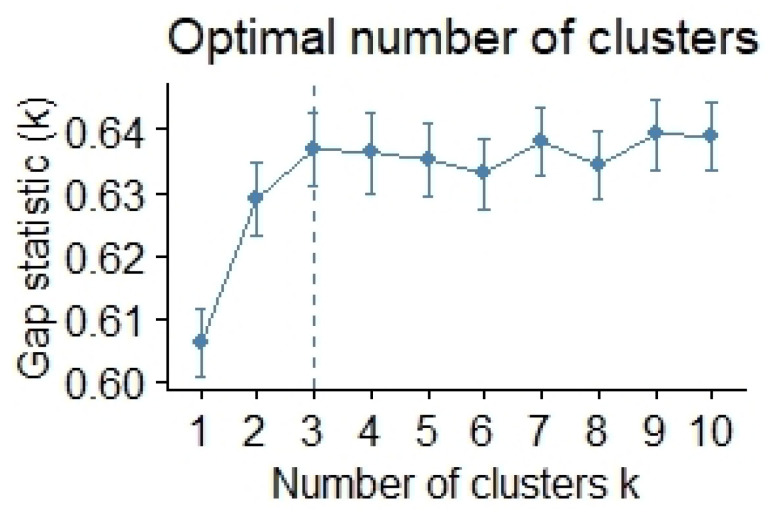
The optimal number of clusters.

**Figure 2 healthcare-10-00888-f002:**
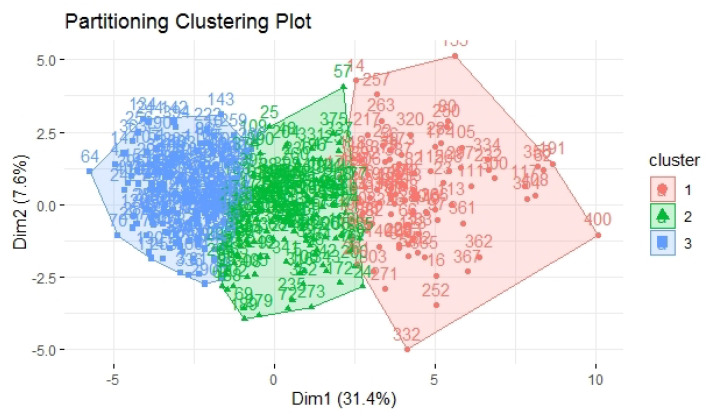
The visualization of the clusters along with members.

**Figure 3 healthcare-10-00888-f003:**
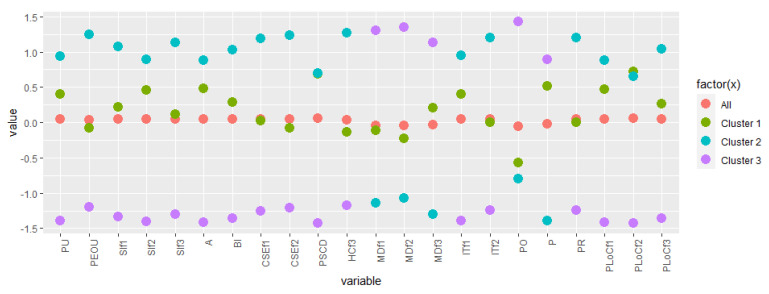
The values for the clusters and the overall population. PU—perceived usefulness; PEOU—perceived ease of use; SI—social influence; A—attitude; BI—behavioural intention; CSE—computer self-efficacy; PSCD—perceived severity of chronic disease; HC—health consciousness; MD—moral disengagement; IT—institutional trust; PO—political orientation; P—privacy; PR—prosocial responsibility; PLoC—perceived locus of causality.

**Figure 4 healthcare-10-00888-f004:**
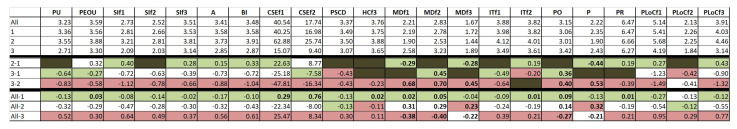
The table describes the differences in the average values of all the variables. For a given variable (a column), green values indicate the highest difference in absolute value, and red values indicate the smallest. Dark brown cells indicate non-meaningful differences for a variable in a pair of clusters.

**Figure 5 healthcare-10-00888-f005:**
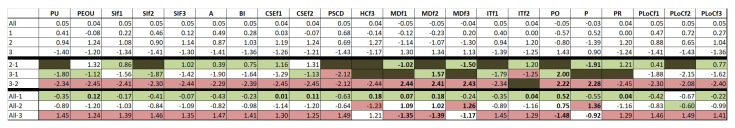
The table describes the scaled differences of the average values of all the variables. For a given variable (a column), green values indicate the highest difference in absolute value, and red values indicate the smallest. Dark brown cells indicate non-meaningful differences for a variable in a pair of clusters.

**Table 1 healthcare-10-00888-t001:** Measures of cluster fitness.

Metrics	Value
km.res$totss	220,847.1
km.res$withinss	[c1 = 20,097.21, c2 = 14,695.99, c3 = 18,055.22]
km.res$tot.withinss	52,848.43
km.res$betweenss	167,998.6
km.res$betweenss/km.res$tot.withinss	3.178876
km.res$size	[c1 = 145, c2 = 141, c3 = 122]
km.res$betweenss/km.res$totss	0.7607012

**Table 2 healthcare-10-00888-t002:** Performance measures to compare the three models.

	Accuracy	AIC	LogLoss
Full model	0.9012346	116	0.3119839
Step-wise reduction model	0.9382716	52.066	0.5360709
Knowledge-based model	0.9876543	48	0.6150691

## Data Availability

The data presented in this study are available on request from the corresponding author. The data are not publicly available due to privacy reasons.

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
