# Peer review of "A Cluster Analysis of the Acceptance of a Contact Tracing App—The Identification of Profiles for the Italian Immuni Contact Tracing App"

_healthcare, 2022, doi:10.3390/healthcare10050888_

Round 1
Reviewer 1 Report
The authors used adequate methodology and included enough details in different sections with summary tables and figures. Below are minor comments for authors to consider.
Can you please elaborate on values you considered for the k-mean clustering algorithm? I believe the values for the clusters (y axis) in the figure 3 is scaled values. For instance, PLoC 7-point scale. Did you scaled to mean 0 and variance 1? I think the k-means clustering were performed with scaled values matrix?
The best way is to treat PU, PEOU and all other variables as categorical variables. However, it’s okay to use it as continuous measures. Before performing ANOVA, we need to check assumptions. Does it satisfy ANOVA assumptions. It would be easy to perform multinomial logistic regression (y= three groups, x= categorical variables/socio- Anagraphical data & continuous data /PU,PEOU, etc). Factors can be selected using backward elimination method.
Line 697 needs to be relaxed. Multinomial logistic regression can be used for multi class problem.
Reviewer 2 Report
This is an interesting paper that focus on different behaviours when adapting and using contact tracing apps. It is very interesting to better understand this issue which was not successful in most of the countries where apps were developed by different government levels (regional or national) were not successful enough.
The paper is well written and structures, however I have some recommendations.
- The abstract does not show results of the paper, it seems more an introduction thant a summary of the content.
- The final paragraph of the introduction is not necessary. It would be great to finish the introduction just with the aim of the study.
- I suggest to add a description of the Immuni App with some information about the kind of interaction this app has with the end-user and how data is stored and integrated in the italian health system.
- I suggest adding the 120 questions in a supplementary file.
- Was the survey piloted before being launched to the community? Did you validate somehow the questionnaire?
- I suggest skipping table 1 as it is not adding value to the information of the paper.
- In the results section, methods are mixed with results. I suggest to better show only results in the Results section and add all the information about how did you perform the analysis in the methodological section.
- Have you considered to reduce the number of items in the questionnaire? Maybe you can do an analysis to identify which are the key questions that are reflected in the 3 different profiles and then a tool to identify citizens profile can be set up and used in future pandemics or crisis situations.
Round 2
Reviewer 2 Report
The paper is ready to be published.